# Impacts of extreme climate events and disturbances on carbon dynamics

**Jingfeng Xiao[1], Shuguang Liu[2], and Paul C. Stoy[3]**

[1]Earth Systems Research Center, Institute for the Study of Earth, Oceans, and Space, University of New Hampshire, Durham, NH 03824, USA

[2]U.S. Geological Survey (USGS) Earth Resources Observation and Science (EROS) Center, Sioux Falls, SD 57198, USA

[3]Department of Land Resources and Environmental Sciences, Montana State University, Bozeman, MT 59717, USA

Correspondence to: J. Xiao (j.xiao@unh.edu)

**Abstract** The impacts of extreme climate events and disturbances (ECE&D) on the carbon cycle have received growing attention in recent years. This special issue showcases a collection of recent advances in understanding the impacts of ECE&D on carbon cycling. Notable advances include quantifying how harvesting activities impact forest structure, carbon pool dynamics, and recovery processes; observed drastic increases of the concentrations of dissolved organic carbon and dissolved methane in thermokarst lakes in western Siberia during a summer warming event; disentangling the roles of herbivores and fire on forest carbon dioxide flux; direct and indirect impacts of fire on the global carbon

balance; and improved atmospheric inversion of regional carbon sources and sinks by
incorporating disturbances. Combined, studies herein indicate several major research needs.
First, disturbances and extreme events can interact with one another, and it is important to
understand their overall impacts and also disentangle their effects on the carbon cycle.
Second, current ecosystem models are not skillful enough to correctly simulate the underlying
processes and impacts of ECE&D (e.g., tree mortality and carbon consequences). Third,
benchmark data characterizing the timing, location, type, and magnitude of disturbances must
be systematically created to improve our ability to quantify carbon dynamics over large areas.
Finally, improving the representation of ECE&D in regional climate/earth system models and
accounting for the resulting feedbacks to climate are essential for understanding the
interactions between climate and ecosystem dynamics.

## 1   Introduction

The biosphere plays an important role in regulating atmospheric carbon dioxide
concentrations and thereby climate. Extreme climate events such as drought (Xiao et al.,
2009; Zhao and Running, 2010) and disturbances such as fire (Bowman et al., 2009),
hurricanes (Chambers et al., 2007; Dahal et al., 2014b; Xiao et al., 2011), wind storms
(McCarthy et al., 2006), and insect outbreaks (Kurz et al., 2008a) can substantially alter
ecosystem structure and function and influence terrestrial carbon dynamics. ECE&D are
projected to increase in both frequency and severity during the remainder of the 21st century
(IPCC, 2013), with important consequences for terrestrial carbon cycling. Projecting the
impacts of these future events remains a challenge given the substantial uncertainty in
forecasting these events and the insufficient representation of ECE&D in ecosystem and land
surface models. A better understanding of the impacts of ECE&D on carbon dynamics across
different ecosystems is essential for projecting ecosystem responses to future climate change
and feedbacks to the climate system.

Biospheric carbon fluxes often exhibit pronounced interannual variability (IAV) and

ECE&D are believed to be primary sources of the IAV (Eimers et al., 2008; Reichstein et al.,
2013; Xiao et al., 2014), which can be pronounced. For example, gross primary productivity
(GPP) exhibited significant IAV over the period 2000-2014 on the global scale as identified
by the MODIS GPP product (Zhao et al., 2005), with important regional differences (Fig. 1).
The IAV is measured by the coefficient of variation (CV), defined as the standard deviation
divided by the mean. Australia and southern Africa had the largest IAV; the U.S. Great
Plains, the U.S. Southwest, Alaska, India, part of the Tibetan Plateau, eastern Mongolia,
Kazakhstan, the Sahel region, and eastern Amazon had intermediate IAV; the remaining
regions had relatively low IAV.

ECE&D can lead to exceptionally high or low annual carbon fluxes. We used the

annual GPP data from the MODIS GPP product (Zhao et al., 2005) to identify extreme GPP
values (outliers) that exceed the statistical normal range presumably caused by extreme
climate events and/or disturbances (Fig. 2). For each grid cell, the outliers of annual GPP over
the period 2000-2014 were identified using interquartile range (IQR) and quartiles (Q1: 25%
quartile; Q3: 75% quartile). The outliers on the higher end were determined as values beyond
$IQR + 1.5 \times Q3$, and the outliers on the lower end were identified as values below $IQR - 1.5$
$\times Q1$. Outliers on the lower end were observed in parts of Europe, Russia, North America, the
Amazonia, and Africa (Fig. 2). These exceptionally low annual GPP were likely caused by
drought, extreme low temperature, fire disturbance, or harvesting. Outliers on the higher end
were observed in Alaska, the U.S. Southwest, Australia, and parts of the Amazonia and
southern Africa (Fig. 2). These exceptionally high annual GPP were likely caused by

exceptionally moist conditions and/or warm temperatures. The U.S. Great Plains and Kazakhstan had large IAV and outliers on the lower end; part of Australia and southern Africa also exhibited large IAV but had outliers on the higher end; the large IAV of GPP did not correspond to outliers for other regions (Figs. 1 and 2). The IAV of carbon fluxes was likely driven by both outliers and moderate to strong anomalies in fluxes.

The impacts of ECE&D on carbon dynamics have received growing attention. We searched the number of journal articles on these topics using Web of Science (Fig. 3) and found a total of 497 and 1593 journal articles for extreme climate events and disturbances, respectively, over the period from 2000 to 2015. Notably, the annual number of publications on the impacts of these events on carbon dynamics has been growing at an average rate of 18 articles per year from 2000 to 2015 and at an average rate of 25 articles per year over the past decade (2006-2015), emphasizing the growing scientific interest in these important topics.

Various approaches have been used to assess the impacts of ECE&D on ecosystem carbon dynamics. At the ecosystem scale, in-situ methods including field experiments (Barbeta et al., 2013), long-term observations (Turner et al., 2003), and the eddy covariance technique (Amiro et al., 2010; Schwalm et al., 2010) seek to understand the mechanisms underlying responses of ecosystem processes to ECE&D. Modeling approaches including process-based ecosystem models (Liu et al., 2011) or data-driven upscaling approaches (Jung et al., 2009; Xiao et al., 2008) have been used for regional to global assessments, which also rely heavily on satellite remote sensing (Xiao et al., 2014). Synthesizing these findings is an ongoing challenge, and multiple approaches are required to understand the consequences of different ECE&D for carbon cycling.

Spatially, the locations of the previous research activities have been largely aligned with the geography of the occurrence of ECE&D. For example, we have witnessed

pronounced impacts of insect outbreaks and fires in the northern Rocky Mountains (Hicke et
al., 2012b; Kurz et al., 2008b; Law et al., 2004), the widespread deforestation in Amazon and
other tropical regions (Achard et al., 2014; DeFries et al., 2002; Harris et al., 2012), peatland
fires in Indonesia (Page et al., 2002; Turetsky et al., 2015), tropical cyclones in the United
States (Dahal et al., 2014a), and drought and heat waves in Europe (Bréda et al., 2006; Ciais
et al., 2005a; Reichstein et al., 2007) and the southwestern United States (Allen et al., 2010a;
Carnicer et al., 2011; Zeppel et al., 2013). Temporally, most of the research has been on the
impacts of individual ECE&D, with fewer studies involving long-term observations and
monitoring records (Dahal et al., 2014a; Seidl et al., 2014). Abundant evidence has been
collected globally in the past decades on increased tree mortality resulting from climate
events such as prolonged mega droughts and heat waves (Allen et al., 2010a; McDowell,
2011; Meddens et al., 2015; Meir et al., 2015). However, the mechanisms behind this
increased mortality and the consequences on carbon dynamics still remain to be unveiled
(Meddens et al., 2015; Meir et al., 2015).
The present special issue is the outcome of special sessions on the impacts of ECE&D
on carbon dynamics at the American Geophysical Union Fall Meeting (2011-2013). This
issue consists of 17 articles: 6 on extreme climate events and 11 on disturbances. This special
issue, along with the special issue on climate extremes and biogeochemical cycles in
*Biogeosciences* (Bahn et al., 2015), reflects recent advances in assessing how ECE&D
influence terrestrial carbon cycling. We feel that the authors have provided a timely and
valuable contribution to the research communities of carbon cycle and global change.
**2    Methods and Findings**
We highlight the findings in this special issue by grouping manuscripts that emphasize the
impacts of drought and extreme precipitation events, herbivory (namely insect outbreaks),
fire, interactions between herbivory and fire, natural hazards (e.g. hurricanes and typhoons),
and forest management.
*Drought and extreme precipitation events*
Piayda et al. (2014) quantified the impacts of the extreme drought event in 2012 on carbon
and water cycling in a Mediterranean woodland. The drought reduced overstory GPP in 2012
by 28% and carbon sink strength by 38% compared to 2011. Results indicated that successful
simulation of drought effects on the montado ecosystem requires the incorporation of variable
apparent maximum carboxylation rate, stomatal conductance, and vapor pressure deficit
sensitivity into photosynthesis-stomatal conductance modeling.

The simulations of a process-based ecosystem model showed that drought from 2000

to 2011 led to significant reduction in both GPP and net ecosystem productivity (NEP) of
China's terrestrial ecosystems at regional to national scales (Liu et al., 2014). Relative to the
long-term mean, the nationwide annual NEP in 2001, 2006, 2009, and 2011 decreased by *ca.*
63, 88, 170, and 61 Tg C yr$^{-1}$, respectively, due to droughts (Liu et al., 2014). These two
studies were consistent with several previous synthesis and modeling studies indicating that
severe droughts could reduce annual GPP and net ecosystem productivity (NEP), and the
reduction in NEP was largely driven by the decrease in GPP due largely to reductions in GPP
(Ciais et al., 2005b; Schwalm et al., 2010; Xiao et al., 2009).

The opposite of drought – extreme precipitation events - have received less attention

in carbon cycle studies. Jiang et al. (2013) conducted a field experiment in three subtropical
forests to study the responses of soil respiration to both drought and extreme high
precipitation and found that altered precipitation strongly influenced soil respiration not only
by controlling soil moisture but also by modifying moisture and temperature sensitivity of soil
respiration. Their results indicate that soil respiration was more sensitive to soil moisture in
the presence of drought, and higher precipitation in the wet season could have a limited effect
on the response of soil respiration to rising temperatures (Jiang et al., 2013).

Zeppel, Wilks, and Lewis (2014) reviewed studies of extreme precipitation and

seasonal changes in precipitation on carbon metabolism in grassland and forested ecosystems.
They found that extremely high precipitation is likely to increase aboveground net primary
productivity (ANPP) of xeric biomes and reduce ANPP of mesic biomes. Changes in
precipitation during the growing season are likely to have a greater impact on carbon cycle
dynamics than precipitation during the non-growing season (Zeppel et al., 2014). These
studies indicated that the direction and magnitude of the impacts of extreme precipitation
events on carbon fluxes depend on the season (wet versus dry) and biome type (xeric versus
mesic).
*Extreme temperature events*
Extreme temperature events have been a feature of recent climate change, especially at high
latitudes (IPCC, 2013). Previous studies showed that extreme temperature events often reduce
GPP and NEP of terrestrial ecosystems (Ciais et al., 2005b; Qu et al., 2016). The effects of
extreme temperature on the carbon dynamics of aquatic ecosystems, however, have received
little attention. Pokrovsky et al. (2013) studied the impacts of the 5 – 15 °C summer warming
event of 2012 on the carbon dynamics of thermokarst lakes in western Siberia. Dissolved
organic carbon concentrations increased by a factor of two as a result of the warming event
despite limited changes in conductivity and pH, and the concentration of dissolved methane
increased by nearly fivefold (Pokrovsky et al., 2013). These results demonstrate a substantial
increase in the methane emission capacity from lakes as a result of summertime warming in
areas of permafrost thaw.
De Simon et al. (2013) examined the effects of manipulated warmer or cooler late
winter/early spring conditions on the carbon budget and yield of soybean crops. Their results
demonstrate that extreme temperature events in late winter did not result in significant
changes in the net carbon balance (De Simon et al., 2013). These events may have larger
impacts on natural ecosystems by advancing or delaying leaf-out dates.
Combined, these studies indicate that the effects of extreme temperature events on
ecosystem carbon dynamics depend on the timing and magnitude of these events. Extreme
temperature events occurring in the growing season could substantially alter carbon fluxes,
while those events occurring during the remainder of the year had smaller effects than
expected.
*Insect outbreaks*
The coniferous forests of western North America have experienced an unprecedented
herbivore outbreak over millions of hectares over the past decades (Hicke et al., 2012a; Raffa
et al., 2008), part of the global tree die-off due to the combined effects of elevated
temperatures, drought, and associated herbivory (Allen et al., 2010b). Measurements of the
impacts of this disturbance at the site scale find minimal ecosystem carbon loss or even net
uptake shortly after eruptive herbivory (Brown et al., 2010), which contrasts regional
estimates of substantial carbon losses to the atmosphere (Ghimire et al., 2015; Kurz et al.,
2008a). Mathys et al. (2013) in this issue used the eddy covariance technique to study carbon
dioxide flux after a mountain pine beetle (*Dendroctonus ponderosae*, Hopkins) attack over a
two-year period and compared these to an adjacent clearcut. They found that the mountain
pine beetle-damaged forest was a carbon sink of *ca.* 50 g C $m^{-2}$ $year^{-1}$ two years after attack.
This study also indicates that the residual forest and the understory vegetation contributed to
carbon uptake and could enable the forest to return to carbon neutrality at a faster rate than
clearcuts. The impacts of herbivore outbreak depend on the type of herbivore (e.g. foliavores
*versus* phloem-feeders) and the intensity of disturbance (Allen et al., 2010b; Brown et al.,
2010; Ghimire et al., 2015; Hicke et al., 2012a; Kurz et al., 2008a; Mathys et al., 2013; Raffa
et al., 2008).
*Fire*
Fire causes direct and immediate carbon emissions into the atmosphere from biomass burning
(the direct effect), and subsequent changes in NEP (the indirect effect) through changes in
GPP and ecosystem respiration of the remaining live stand and the heterotrophic respiration
of the damaged biomass. Li et al. (2014) in this special issue provided a quantitative
assessment of the direct and indirect impacts of fire on the net carbon balance of global
terrestrial ecosystems during the 20th century. Their results show that fire decreased the net
carbon gain of global terrestrial ecosystems by 1.0 Pg C yr$^{-1}$ averaged across the 20th century,
as a result of the fire direct effect (1.9 Pg C yr$^{-1}$) partly offset by the indirect effect ($-0.9$ Pg C
yr$^{-1}$). The effect of fire on the net carbon balance significantly declined until 1970 with a
trend of 8 Tg C yr$^{-1}$ due to an increasing indirect effect, and increased subsequently with a
trend of 18 Tg C yr$^{-1}$ due to an increasing direct effect (Li et al., 2014). These results help
constrain the global-scale dynamics of fire and the terrestrial carbon cycle.
*Insect outbreaks* versus *fire*
At the regional scale, Caldwell et al. (2013) simulated and evaluated the long-term impacts of
the two characteristic disturbances in the Southern Rocky Mountains forests (i.e., the outbreak
of mountain pine beetle and high-severity wildfire) on changes in species composition and
carbon stocks. Wildfire caused larger changes in both patterns of succession and distribution
of carbon among biomass pools than did mountain pine beetle disturbance; carbon in standing
live biomass returned to pre-disturbance levels after 50 versus 40 years following wildfire and
mountain pine beetle disturbances, respectively (Caldwell et al., 2013).

Clark et al. (2014) used the eddy covariance technique to study the impacts of fire and

gypsy moth (*Lymantria dispar* L.) disturbance in oak-dominated, pine-dominated, and mixed
forests in eastern North America. The net ecosystem exchange (NEE), GPP, and water use
efficiency were of greater magnitude in the oak-dominated forest before disturbance during
summer. Water use efficiency declined by 60% at the oak-dominated stand and by nearly
50% at the mixed stand after gypsy moth disturbance, but prescribed fire had little impact on
water use efficiency in the mixed or pine stands (Clark et al., 2014). These results
demonstrate the importance of forest type, disturbance type, and time since disturbance on
coupled carbon and water cycle functioning in temperate forests.
*Hurricanes and typhoons*
Hurricane events in the U.S. have significant effects on regional carbon dynamics (Dahal et
al., 2014b). Typhoons are natural disturbances to subtropical mangrove forests in Asia, and
their effects on ecosystem carbon dynamics of mangroves are not well understood. Chen et al.
(2014) examined the short-term effects of frequent strong typhoons on defoliation and the
NEE of subtropical mangroves. The responses of daily NEE following typhoons were highly
variable in mangrove ecosystems (Chen et al., 2014), demonstrating that the characteristics of
the typhoon and antecedent ecosystem conditions are important for understanding hurricane
impacts on carbon stocks and fluxes. Severe hurricanes and typhoons that destroy a large
number of trees could have significant effects on regional carbon cycling, while those that
lead merely to defoliation likely had transient effects on ecosystem carbon exchange.
*Forest management*
Accurate quantification of the effects of partial cutting or clearcutting is essential for a better
understanding of forest carbon dynamics and for informing forest management. Zhou et al.
(2013a) conducted a meta-analysis on the impacts of partial cutting (i.e., cutting events with
aboveground biomass removal rate < 90%) on forest carbon stocks by collecting data on
cutting intensity, forest structure, and carbon stock components. This is a global-scale meta-
analysis, but the majority of the sites are distributed in the U.S. and Europe. The results
showed that partial cutting reduced aboveground carbon by 43% and increased understory
carbon storage by nearly 400% on average, but did not have significant effects on forest floor
or mineral soil carbon stocks (Zhou et al., 2013a). This effort provides a new perspective on
the impacts of forest harvesting as it covers the spectrum of harvest disturbances from partial
cutting to clearcut and goes beyond previous reviews that mostly concentrated on the impacts
of clearcutting (Johnson and Curtis, 2001; Nave et al., 2010). The impacts of partial cutting
can be significant; for example, partial cutting accounted for about three quarters of the total
C loss from timber harvesting in the eastern United States from 2002 to 2010 (Zhou et al.,
2013b).

Wang et al. (2014) used a process-based forest ecosystem model, PnET-CN, to

evaluate how clearcutting alters ecosystem carbon fluxes, biomass, and leaf area index in
northern temperate forests. They found that harvest disturbance in northern temperate forests
had significant effects on forest carbon fluxes and stocks, and increased harvesting intensity
would delay the recovery of NEP. Evergreen needleleaf forests were slower to recover to full
carbon assimilation capacity after stand-replacing harvests than deciduous broadleaf forests
(Wang et al., 2014). Future modeling studies of disturbance effects should incorporate forest
population dynamics (e.g., regeneration and mortality) and relationships between age-related
model parameters and state variables (e.g., leaf area index).
*Disturbance legacy*
The time since disturbance is an important controlling factor of carbon dynamics. Berryman
et al. (2013) tested the impacts of experimental pinyon pine (*Pinus edulis* Englem.) mortality
on microbial respiration. They found that litter respiration responded to water availability at
both treatment and control sites, and that soil respiration decreased at the site with
experimental mortality. These results demonstrate ecosystem-level consequences of tree
mortality that differs as a function of water availability (Berryman et al., 2013).

Yue et al. (2013) compared observations from post-fire vegetation trajectories in the

boreal forest with simulations from the process-based ORCHIDEE vegetation model and
supported the notion that the increase in atmospheric $CO_2$ concentrations and vegetation
recovery were jointly responsible for current carbon sink conditions. It should be noted that
nitrogen deposition – a global change factor enhancing ecosystem carbon uptake was not
explicitly considered, although the effects of nitrogen deposition carbon sink strength have
been controversial (Magnani et al., 2007; Nadelhoffer et al., 1999). Nevertheless, their results
highlight the importance of understanding how global change and disturbance events interact
to determine current – and likely future – carbon cycle dynamics (Yue et al., 2013). These
two studies demonstrate that the legacy of disturbance and environmental factors jointly
control the carbon dynamics following disturbance.

Modeling approaches have been widely used to simulate ecosystem carbon dynamics

following disturbance. Wang et al. (2014) in this special issue simulated the dynamics of
carbon fluxes and stocks following harvest. The simulated NEP and aboveground carbon
stock after clearcuts generally followed the hypothesized trajectories (Chapin, 2011; Odum,
1969) while the decline in NEP was due to relatively stable GPP and gradually increasing
ecosystem respiration (ER). Evergreen needleleaf forests recovered more slowly from a net
carbon source to a net sink, and lost more carbon than deciduous broadleaf forests.

Disturbance-induced tree mortality regulates the forest carbon balance, but tree

mortality and its carbon consequences are not well represented in ecosystem models (Bond-
Lamberty et al., 2015). Bond-Lamberty et al. (2015) tested whether three ecosystem models –
the classic big-leaf model Biome-BGC and the gap-oriented models ZELIG and ED - could
reproduce the resilience of forest ecosystems to moderate disturbances. The models replicated
observed declines in aboveground biomass well but could not fully capture observed post-
disturbance carbon fluxes. This study indicates that ecosystem models are yet unable to
correctly simulate the effects of disturbances.

Lack of critical geospatial data on disturbances and associated impacts on ecosystems

has been identified as one of the main challenges in quantifying carbon dynamics over large
areas (Liu et al., 2011). Recently, a continental-scale forest stand age map was developed for
North America using forest inventory data, large fire data, and remotely sensed data,
providing a new source of information that can benefit quantification of the carbon sources
and sinks across the continent and contribute to studies of disturbance (Pan et al., 2011). Deng
et al. (2013) in this special issue used these continental stand age maps as an additional
constraint to atmospheric $CO_2$ inversions. They found that regions with recently disturbed or
old forests are often nudged towards carbon sources while regions with middle-aged
productive forests are shifted towards sinks, confirming stand age effects observed from many
eddy covariance flux towers (Deng et al., 2013). These results were generally consistent with
the synthesis results from eddy covariance flux data across North America (Amiro et al.,
2010) but they were inconsistent with some other studies showing that old-growth forests
were still carbon sinks (Desai et al., 2005; Luyssaert et al., 2008). At the sub-continental
level, their inverted carbon fluxes agreed well with continuous estimates of NEE upscaled
from eddy covariance flux data (Xiao et al., 2008; 2011). Recent development in
characterizing the timing, location, type, and magnitude of disturbances (Huang et al., 2010;
Kennedy et al., 2010; Masek et al., 2013; Williams et al., 2014; Zhu and Woodcock, 2014)
are helping to advance diagnosis and monitoring of carbon dynamics over large areas.

## 3   Conclusions

The contributions of this special issue reflect some of the most recent advances in the impacts of ECE&D on carbon dynamics. These studies address the impacts of different types of extreme events including forest management, hurricanes and typhoons, drought, extreme precipitation events, extreme temperature events, insect outbreaks, and fire as well as ecosystem recovery since disturbance. The direction and magnitude of the effects of these events on ecosystem carbon fluxes depend on the nature of the events (type, duration, and intensity), the timing of the events (e.g., wet versus dry season, summer versus winter), and the biome type (e.g., xeric versus mesic). These events typically have negative effects on net carbon uptake while some events such as extreme precipitation events may also have positive effects on net carbon uptake depending on antecedent conditions and the nature of the extreme events.

Importantly, studies in this special issue collectively indicate several major research needs. First, ECE&D can interact with one another, and it is important to disentangle their relative effects on the carbon cycle. Second, current ecosystem models in general are not skillful enough to correctly simulate the impacts of ECE&D such as disturbance-induced tree mortality and its carbon consequences, and therefore ecosystem models must be improved to correctly represent the underlying processes and impacts (Liu et al., 2011; Reichstein et al., 2013). For example, the processes of drought effects on ecosystem respiration are not well represented in models. Third, the lack of data on major disturbances is still one of the main challenges that hinder the improvement of quantifying carbon dynamics over large areas, and benchmark data characterizing the timing, location, type, and magnitude of disturbances must be created. With the ongoing continuous monitoring of earth surface conditions using a constellation of satellites and emerging data mining technologies, the characterization and

understanding of the impacts of ECE&D are expected to improve drastically over the next 5
to 10 years. However, major challenges still remain on how to translate those conditional
changes into carbon fluxes and understand the specific roles of ECE&D in particular. Finally,
besides carbon fluxes and stocks, other biogeophysical properties such as albedo,
evapotranspiration (ET), and surface energy exchange are also altered by ECE&D. Improving
the representation of ECE&D in regional climate/earth system models and accounting for the
resulting feedbacks to the climate are essential for understanding the interactions between
climate and ecosystem dynamics. Ongoing research in these areas will continue to improve
our emerging understanding of the impacts of ECE&D on carbon cycling and the feedbacks to
the climate.

**Acknowledgements**
We would like to thank all of the scientists who contributed to this special issue. JX
acknowledges support from the National Science Foundation (NSF) through the Emerging
Frontiers Macrosystems Biology Program (award numbers 1065777) and the National
Aeronautics and Space Administration (NASA) through the Carbon Cycle Science Program
(award number NNX14AJ18G) and the Terrestrial Ecology Program (award number
NNX12AK56G). PCS acknowledges the support of NSF Macrosystems Biology Program
(award number 1241810), the NSF Division of Environmental Biology (award number
1552976) and the Alexander von Humboldt-Foundation. SL acknowledges the support from
the U.S. Geological Survey Land Change Science Program. Any use of trade, firm, or product
names is for descriptive purposes only and does not imply endorsement by the U.S.
Government. We thank Dr. M. van Oijen and an anonymous reviewer for their constructive
comments on the manuscript.

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

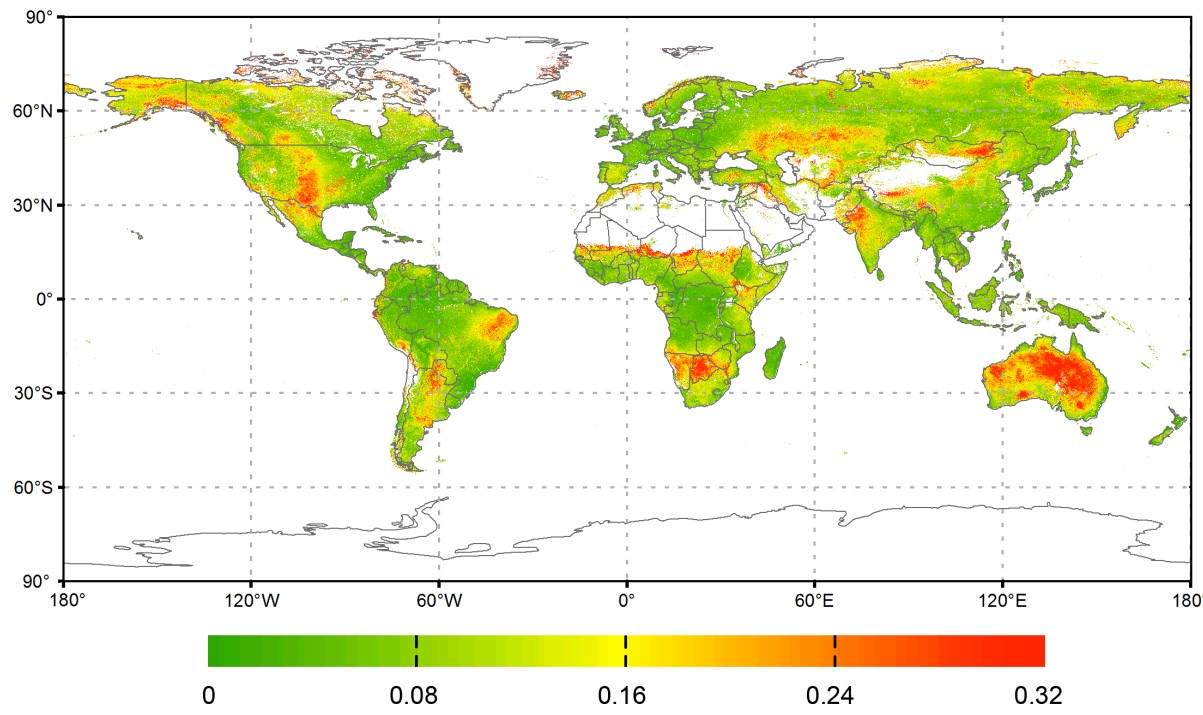


**Fig. 1.** The interannual variability (i.e., the coefficient of variation or CV) of annual gross primary productivity (GPP) over the period 2000-2014 from the MODIS GPP product (MOD17A3). The CV is unitless.


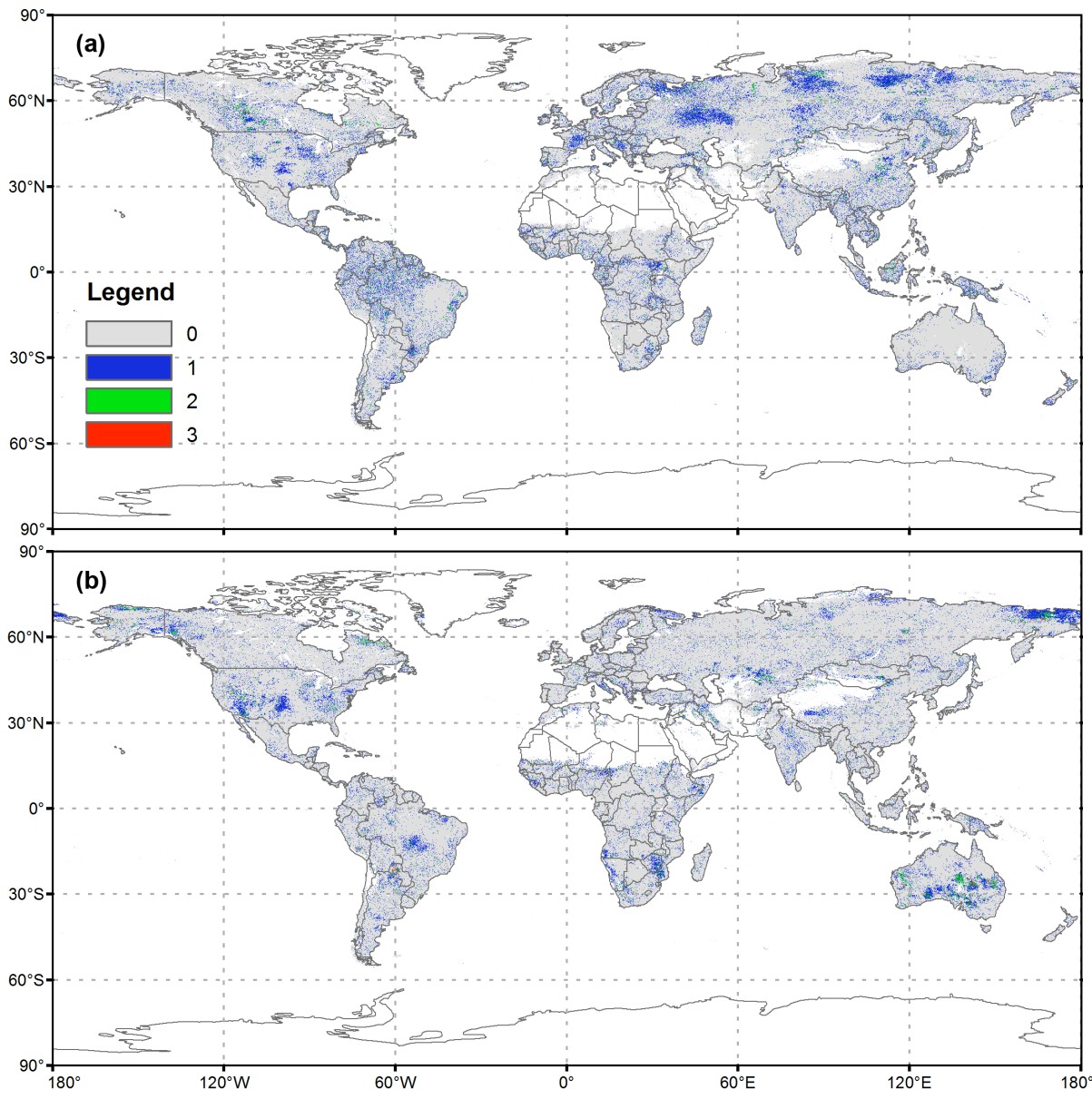


**Fig. 2.** The number of extreme years characterized by the number of outliers of the annual gross primary productivity (GPP) distribution over the period 2000-2014: (a) outliers on the lower end (i.e., exceptionally low annual GPP); (b) outliers on the higher end (i.e., exceptionally high annual GPP). The outliers were identified using interquartile range (IQR) and quartiles (Q1: 25% quartile; Q3: 75% quartile) with the outliers on the higher end determined as values beyond IQR + 1.5 × Q3 and the outliers on the lower end determined as values below IQR − 1.5 × Q1. The annual GPP values (g C m$^{-2}$ year$^{-1}$) were derived from the MODIS GPP product (MOD17A3).

648

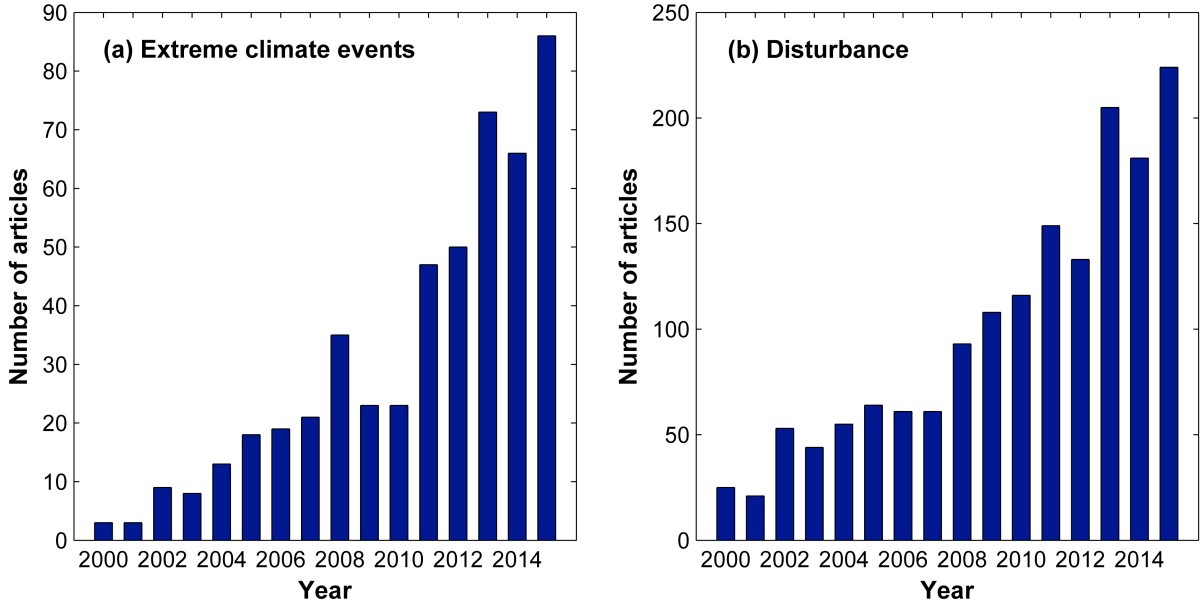

649

Fig. 3. The number of journal articles published over the period from 2000 to 2015 as identified by Web of Science™ as of April 14, 2016 for the impacts of (a) extreme climate events and (b) disturbance on carbon dynamics. The combination of key words that we used to represent 'extreme climate events' is: TS=("extreme climate events" OR "climate extremes" OR drought OR "extreme precipitation") AND TS=("carbon dynamics" OR "carbon cycle" OR "carbon flux" OR "carbon stock" OR "carbon pool"), where TS stands for Topic. The combination of key words used to represent 'disturbance' is: TS=(disturbance OR fire OR harvesting OR logging OR hurricane or "insect outbreaks") AND TS=("carbon dynamics" OR "carbon cycle" OR "carbon flux" OR "carbon stock" OR "carbon pool").