# Peer review of "Impacts of extreme climate events and disturbances on"

_Biogeosciences, 2016_

## Referee Comment (RC1) · Anonymous Referee #1 · 22 Feb 2016

This is a good preface to a great special issue. It might be a bit longer than needed and I'd recommend cutting it down in a few places to provide only a brief, broad framing for the issue and then quickly getting to the narrative. You do a nice job of organizing and connecting the works with a set of succinct highlights of each paper's contribution. The writing could be pithier, more precise, and clearer in a few places here and there. A number of suggestions are offered below that may help.

L16: suggest cutting "as evidenced by the ....topics."

L17, L19: replace "advancements" with "advances"

L19: cut ", but are not limited to,"

L27: awkward, ambiguous wording: "disentangle their relative effects" to "disentangle

their [separate and combined] effects"

L30 to 33: wordy, should be stated more succinctly

L 36 to 80: I found the use of IAV in MODIS GPP to be a somewhat awkward fit for the papers in the issue. Furthermore, I suggest reducing much of the text from lines 36 to 80, and quickly getting to the content of the present special issue.

L 41: "Some tropical regions (ie...)" is awkward, maybe "IAV is particularly high in tropical regions such as ..."

L 62: This is a fairly general set up and is not specific to the papers of this issue. You might cut or shorten this section, not because it's incorrect or irrelevant but only because a special issue preview might be best to quickly get to the review of the papers therein.

L 64: "long-term observations" is vague, lacks a citation as example, and the sentence structure suggests that EC is long-term.

L 65: Why cite Dong et al. 2011? This seems unrepresentative.

L 89: poor sentence structure, maybe "We highlight the findings in this special issue by grouping manuscripts that emphasize ..."

L92 to 96: This is somewhat awkward, almost seeming to undermine the usefulness of the works that are presented. I'd recommend saving the comment about need for work on interactive effects for the discussion of future research directions. Also, L92-93 seems redundant with L27, and has the same issue raised above regarding "relative effects".

L 110: extreme low precipitation is a key facet of drought, not its opposite. Should this be modified to read "extreme [high] precipitation..." ?

L 111 to 115: The setup to this paper's highlight seems to suggest that the study focuses on non-drought conditions. Why then does Line 114 note that soil respiration

would decrease if soil moisture continued to decrease? The narrative reasoning is incongruous here and should be fixed.

L 120: Replace "positive" and "negative" with something clearer. What is a "positive" response of a biome? Is it higher GPP, higher Respiration, higher NEP, higher biodiversity?

L 139 to 143: This statement does not seem to be justified. Winter and spring are not key seasons for metabolic activity in irrigated croplands so the leading statement about smaller effects on the overall annual carbon balance seems to be misleading.

L 143 to 144: "Combined..." This comment about the importance of timing and magnitude does not appear to be a synthesis statement, pertaining to only one study of those highlighted in the special issue.

L 145 to 147: "[However], extreme temperature events occurr[ing] in the growing season could substantially alter carbon fluxes, while those events occurr[ing] during ..."

L 145 to 147: This statement seems to correct or more correctly state the one above (L139 to 143).

L 155: Include citation to: Ghimire B, Williams CA, Collatz GJ, Vanderhoof M, Rogan J, Kulakowski D, Masek JG (2015) "Large Carbon Release from Bark Beetle Outbreaks across Western United States Imposes Climate Feedback", Global Change Biology, doi: 10.11/gcb.12933.

L 159 to 160: clarify "benefit of herbivory to undamaged trees" and also, does this include understory non-tree species?

L 161 to 163: It seems the study highlighted here only looked at MPB and if so, how could it suggest that the impacts of herbivore outbreak depend on the type of herbivore?

L 166: It seems redundant to include NEP and "carbon exchange between the land

and the atmosphere" given that NEP typically includes CO2 and that non CO2 carbon-containing molecules are rarely emphasized and do not seem to have been emphasized in the studies included in this special issue.

L 166: It might make sense to clarify what is meant by "subsequent changes in NEP" by noting the relevant processes such as respiration of disturbance-killed biomass, and any changes to net primary productivity.

L 173 to 174: Check the units on your trend, which should be Tg C yr-1 yr-1... 8 to 18 Tg C yr-1 is pretty big. Should this be over an interval of time?

L 207: The geographic domain of the Zhou et al. study should be reported. Was it global? Was it in North America or Europe? The Amazon? The quantitative figures reported must be region specific.

L 220 to 223: this statement is very general and does not offer much in the way of findings.

L 222: "vulnerable" seems to be an odd term. All forests would be vulnerable only some are targeted because of economic value and modes of production.

L 224: This heading "time since disturbance" does not appear to be a good fit for the studies highlighted below. You might think about a different heading / grouping.

L 228: "near the site" is vague and unclear.

L 230: This paper does not seem to belong under the heading "Time since disturbance". Can it be better linked to the flow of the preview?

L 232: Replace "found" with "supported the notion that". This is not a new finding, really, and is model based, so it seems somewhat out of place to state that it was "found".

L 234: Maybe connect these sentences... "carbon sink conditions, highlighting the importance of ..."

L 239 to 245: suggest cutting this paragraph. It seems out of place and is redundant with things already mentioned elsewhere, including an earlier highlight of the Wang et al. 2014 study. It has a discussion of its own with citations to works outside of the scope of the special issue and thus seems out of place.

L 252: which two? Wang et al. is not described as supporting this statement, so the statement seems to apply only to the Bond-Lamberty et al. 2015 study.

L 256: Should there be a new heading here? Maybe "Challenges and Opportunities"?

L 265: "conforming" to "confirming"

L 274; "will likely help" to "are helping to", and cite (e.g. Williams et al. 2014). Williams CA, Collatz GJ, Masek J, Huang C, Goward S (2014) "Impacts of disturbance history on forest carbon stocks and fluxes: Merging satellite disturbance mapping with forest inventory data in a carbon cycle model framework", Remote Sensing of Environment, 151:57-71, http://dx.doi.org/10.1016/j.rse.2013.10.034.

---

## Referee Comment (RC2) · M. van Oijen (Referee) · 26 Feb 2016

**GENERAL COMMENTS**

This paper provides an introduction to a special issue of Biogeochemistry. The issue consists of 17 papers on the impact of extreme climatic events and disturbances on ecosystem carbon dynamics. Fifteen of the studies are on terrestrial ecosystems, one study is on mangroves and one on lakes. The papers differ strongly in their choice of ecosystem, research question and methodology. That raises the question: what is the purpose of providing an introduction to such a heterogeneous collection? The authors show (Fig. 2) that more than 200 papers are now being published each year on the response of carbon dynamics to extreme events and disturbances, so why do the 17 issue papers merit special attention? The obvious justification for such an introduction

is that it provides an opportunity to place recent papers in context, i.e. review the state of the art and identify remaining research gaps. This is attempted in the paper but could be done more systematically, as discussed in the following.

The first section of the paper ("Introduction") can be summarised as follows: (1) Inter-annual variability (IAV) of GPP is especially large in the tropics, (2) extreme events and disturbances can affect carbon dynamics and will become more frequent and intense in the future, (3) their impacts can be studied with measurements and models - and many papers are being published; the ones here were the outcome from AGU-sessions.

This is fine as far as it goes, with the possible exception of the text on IAV measured as multi-annual standard deviation of GPP (Fig. 1), which seems an unnecessary distraction. Extreme events are outliers, not standard deviations, and occur worldwide - and not just in the tropics.

A brief analysis of the existing literature could be added to this introductory section, flagging up both what research has been done and what has not been done. Which ecosystem types, which extreme event types, which disturbances, at which locations and spatiotemporal scales have been studied so far, and which were overlooked? And which are likely to become more important in the future? Can we distinguish direct and indirect effects of disturbances and extreme events both on the same location and elsewhere? Do the 17 papers address any of the research gaps? I think the paper would gain from being more comprehensive and analytical - otherwise there is no added value compared to the special issue papers themselves. I understand that you want to keep the preface short, but you could delete the IAV-text and figure, replacing it with say 20-30 lines on the state of the art.

The second section of the paper consists of short summaries of the findings of the 17 papers. These are well-written and will be helpful to those readers who do not want to read the abstracts of the original papers.

The final section ("Conclusions") states future research needs. Three topics are mentioned: (1) studying interactions between extreme events and disturbances, (2) collecting more data on disturbances, (3) improving models for disturbances. Whilst these certainly constitute worthwhile efforts, they seem an arbitrary and small selection of topics; many others could have been mentioned. And are there no research needs left for extreme events rather than disturbances? Also, there is no discernible relationship between the three listed research needs and the 17 papers of the special issue, so it remains unclear what the papers collectively have contributed. For example, at least five of the 17 papers used models: if those models still need to be improved, does that disqualify their current results?

SPECIFIC COMMENTS

l. 36-37: The Introduction begins with discussion of "terrestrial biosphere" and "Terrestrial carbon fluxes". This suggests that the special issue only considers terrestrial ecosystems, which is not the case. Begin by setting the scene (what kind of studies are being introduced by you) before delving into details like the IAV. l. 41: Add a reference to the MODIS work. l. 99-100: Those reductions of 28 and 38% are for which period? During the event, the year following the event, . . .? l. 172-174: What happened around 1970 that caused the trend break? l. 215-216: Here you explain what partial cutting is, after having discussed the impacts of it already on l. 207. l. 231-238: This section seems to ignore the current understanding that it is increased N-deposition, not elevated $CO_2$, that has increased forest sink strength. l. 252: Which two studies?

TECHNICAL CORRECTIONS

l. 24: Missing space after "by". l. 28: Remove "layers". l. 42: "the Amazon" should be "Amazonia". l. 43, 45, 46: Remove "on the order". l. 50-51: Remove "terrestrial"? l. 59-61: "We can only . . . scales" can safely be removed. l. 65: Replace "mechanistic responses" by "mechanisms underlying responses". l. 71: Add "the" before "consequences". l. 76: The total number increases by 200 articles per year, not 20. Replace "total" with "annual". l. 82: AGU meeting: in which year(s)? l. 85-87: "We feel . . .

change" can be removed. l. 93-96: "That being said, . . . 2008": more waffling, please remove. l. 110: Replace "have" with "has". l. 145-146: replace "occurred" with "occurring". l. 179: Remove "potential". l. 182: Replace first dash with a space. l. 201: Write "hurricanes". l. 203: Remove "annual". l. 256: Why write "data layers" instead of simply "data"? There is some GIS-jargon here (including the "polygons" of line 259 and two further "layers" in lines 290 and 292). l. 260: "source of information". l. 265: What does "conforming" mean? l. 293: Remove "systematically".

---

## Author Comment (AC2) · 13 May 2016

**Response to Referee #2's comments**

In preparing this revision, we have fully considered the reviewer (Dr. M. van Oijen)' comments and have revised the manuscript accordingly.

This paper provides an introduction to a special issue of Biogeochemistry. The issue consists of 17 papers on the impact of extreme climatic events and disturbances on ecosystem carbon dynamics. Fifteen of the studies are on terrestrial ecosystems, one study is on mangroves and one on lakes. The papers differ strongly in their choice of ecosystem, research question and methodology. That raises the question: what is the purpose of providing an introduction to such a heterogeneous collection? The authors show (Fig. 2) that more than 200 papers are now being published each year on the response of carbon dynamics to extreme events and disturbances, so why do the 17 issue papers merit special attention? The obvious justification for such an introduction is that it provides an opportunity to place recent papers in context, i.e. review the state of the art and identify remaining research gaps. This is attempted in the paper but could be done more systematically, as discussed in the following.

We thank the reviewer for the constructive comments on our preface. We have revised the manuscript as suggested.

The first section of the paper ("Introduction") can be summarised as follows: (1) Interannual variability (IAV) of GPP is especially large in the tropics, (2) extreme events and disturbances can affect carbon dynamics and will become more frequent and intense in the future, (3) their impacts can be studied with measurements and models - and many papers are being published; the ones here were the outcome from AGU-sessions.

This is fine as far as it goes, with the possible exception of the text on IAV measured as multi-annual standard deviation of GPP (Fig. 1), which seems an unnecessary distraction. Extreme events are outliers, not standard deviations, and occur worldwide - and not just in the tropics.

We have replaced the interannual variability (IAV) map measured by standard deviation with the map measured by the coefficient of variation (CV; CV=standard deviation/mean). CV betters measures the IAV of carbon fluxes. We have also added a figure of the number of extreme annual values (i.e., outliers) (listed as Fig. 2). We identified the outliers on a per-pixel basis using the Boxplot concept. An outlier is defined as an annual GPP value that is either larger than the 75% quartile+1.5*interquartile range or smaller than the 25% quartile – 1.5 * interquartile range.

A brief analysis of the existing literature could be added to this introductory section, flagging up both what research has been done and what has not been done. Which ecosystem types, which extreme event types, which disturbances, at which locations and spatiotemporal scales have been studied so far, and which were overlooked? And which are likely to become more important in the future? Can we distinguish direct and indirect effects of disturbances and extreme events both on the same location and elsewhere? Do the 17 papers address any of the research gaps? I think the paper would gain from being more comprehensive and analytical - otherwise there is no added value compared to the special issue papers themselves. I understand that you want to keep the preface short, but you could delete the IAV-text and figure, replacing it with say 20-30 lines on the state of the art.

We agree that an overview of the literature would be of interest but, as noted in Figure 3 (originally listed as Figure 2), a large number of manuscripts have been published on the topic of

'extreme events' and on 'disturbances'. Any brief summary of this body of literature would barely do it justice. We feel that in this instance, a full review paper may be able to adequately (but probably not comprehensively) synthesize existing literature. We highlight the novel findings of the manuscripts - this is designed to communicate the gaps in knowledge addressed by each. Instead, we improved Figure 1 to also include an outlier analysis as an alternate approach for identifying regions of the globe that are prone to annual GPP values that exceed the normal statistical range.

The final section ("Conclusions") states future research needs. Three topics are men-tioned: (1) studying interactions between extreme events and disturbances, (2) collecting more data on disturbances, (3) improving models for disturbances. Whilst these certainly constitute worthwhile efforts, they seem an arbitrary and small selection of topics; many others could have been mentioned. And are there no research needs left for extreme events rather than disturbances? Also, there is no discernible relationship between the three listed research needs and the 17 papers of the special issue, so it remains unclear what the papers collectively have contributed. For example, at least five of the 17 papers used models: if those models still need to be improved, does that disqualify their current results?
We have revised this section by adding research needs for extreme events. The need for further improve does not disqualify the current results of these models but indicates our understanding of the underlying mechanisms of extreme climate events and disturbances and their representation in models are still limited.

SPECIFIC COMMENTS

l. 36-37: The Introduction begins with discussion of "terrestrial biosphere" and "Terrestrial carbon fluxes". This suggests that the special issue only considers terrestrial ecosystems, which is not the case. Begin by setting the scene (what kind of studies are being introduced by you) before delving into details like the IAV.
We have removed "terrestrial", and changed "terrestrial carbon fluxes" to "biospheric carbon fluxes".

l. 41: Add a reference to the MODIS work.
We have added a reference for the MODIS data product (Zhao et al. 2005).

l. 99-100: Those reductions of 28 and 38% are for which period? During the event, the year following the event, : : :?
The drought reduced GPP and carbon sink by 28% and 38% in the drought year – 2012. We have clarified this.

l. 172-174: What happened around 1970 that caused the trend break?
l. 215-216: Here you explain what partial cutting is, after having discussed the impacts of it already on l. 207.
We have moved the explanation of partial cutting to where the phrase first appeared in the paragraph.

l. 231-238: This section seems to ignore the current understanding that it is increased N-deposition, not elevated CO2, that has increased forest sink strength.

We have explicitly mentioned that N deposition – a factor enhancing ecosystem carbon uptake was not explicitly considered, although the effects of nitrogen deposition carbon sink strength have been controversial (Magnani et al., 2007; Nadelhoffer et al., 1999).

l. 252: Which two studies?

We have clarified what this study only applies to Bond-Lamberty et al. 2015.

TECHNICAL CORRECTIONS

l. 24: Missing space after "by".
l. 28: Remove "layers".

We have made these changes.

l. 42: "the Amazon" should be "Amazonia".
l. 43, 45, 46: Remove "on the order".
l. 50-51: Remove "terrestrial"?
l. 59-61: "We can only : : : scales" can safely be removed.
l. 65: Replace "mechanistic responses" by "mechanisms underlying responses".
l. 71: Add "the" before "consequences".
l. 76: The total number increases by 200 articles per year, not 20. Replace "total" with "annual".

We have made these changes.

l. 82: AGU meeting: in which year(s)?

We have clarified that the AGU meeting was in 2011-2013.

l. 85-87: "We feel : : : change" can be removed.

We have retained this sentence as part of our evaluation of the authors' contribution..

l. 93-96: "That being said, : : : 2008": more waffling, please remove.
l. 110: Replace "have" with "has".
l. 145-146: replace "occurred" with "occurring".
l. 179: Remove "potential".
l. 182: Replace first dash with a space.
l. 201: Write "hurricanes".
l. 203: Remove "annual".
l. 256: Why write "data layers" instead of simply "data"? There is some GIS-jargon here (including the "polygons" of line 259 and two further "layers" in lines 290 and 292).
l. 260: "source of information".
l. 265: What does "conforming" mean?
l. 293: Remove "systematically".

These changes have been made as suggested.

---

## Author Response (AR1)

**Response to Referee #1's comments**

In preparing this revision, we have fully considered the reviewer' comments and have revised the manuscript accordingly.

L16: suggest cutting "as evidenced by the ....topics."

L17, L19: replace "advancements" with "advances"

L19: cut ", but are not limited to,"

L27: awkward, ambiguous wording: "disentangle their relative effects" to "disentangle their [separate and combined] effects"

We have made these changes as suggested by the reviewer.

L30 to 33: wordy, should be stated more succinctly We have rephrased this sentence for succinctness.

L 36 to 80: I found the use of IAV in MODIS GPP to be a somewhat awkward fit for the papers in the issue. Furthermore, I suggest reducing much of the text from lines 36 to 80, and quickly getting to the content of the present special issue.

We have replaced the interannual variability (IAV) map measured by standard deviation with the map measured by the coefficient of variation (CV; CV=standard deviation/mean). CV betters measures the IAV of carbon fluxes. We have also added a figure of the number of extreme annual values (i.e., outliers) (listed as Fig. 2). We identified the outliers on a per-pixel basis using the Boxplot concept. An outlier is defined as an annual GPP value that is either larger than the 75% quartile+1.5\*interquartile range or smaller than the 25% quartile – 1.5 \* interquartile range.

L 41: "Some tropical regions (ie...)" is awkward, maybe "IAV is particularly high in tropical regions such as ..."

This sentence has been removed because of the replacement of standard deviation with coefficient of variation (CV).

L 62: This is a fairly general set up and is not specific to the papers of this issue. You might cut or shorten this section, not because it's incorrect or irrelevant but only because a special issue preview might be best to quickly get to the review of the papers therein.

We have retained this paragraph as part of the brief summary of the literature on this topic.

L 64: "long-term observations" is vague, lacks a citation as example, and the sentence structure suggests that EC is long-term.

We added a citation for "long-term observations": Turner, M. G., Collins, S. L., Lugo, A. E., Magnuson, J. J., Rupp, T. S., and Swanson, F. J.: Disturbance dynamics and ecological response: The contribution of long-term ecological research, Bioscience, 53, 46-56.

L 65: Why cite Dong et al. 2011? This seems unrepresentative. We replaced this reference with a global-scale cross-site synthesis study (Schwalm et al. 2010). L 89: poor sentence structure, maybe "We highlight the findings in this special issue by grouping" manuscripts that emphasize ..."

We have rephrased the sentence as suggested.

L92 to 96: This is somewhat awkward, almost seeming to undermine the usefulness of the works that are presented. I'd recommend saving the comment about need for work on interactive effects for the discussion of future research directions. Also, L92-93 seems redundant with L27, and has the same issue raised above regarding "relative effects".

We have removed these two sentences.

L 110: extreme low precipitation is a key facet of drought, not its opposite. Should this be modified to read "extreme [high] precipitation..."?

Extreme precipitation events typically refer to exceptionally high precipitation events, and therefore we keep the use of "extreme precipitation events".

L 111 to 115: The setup to this paper's highlight seems to suggest that the study focuses on nondrought conditions. Why then does Line 114 note that soil respiration would decrease if soil moisture continued to decrease? The narrative reasoning is incongruous here and should be fixed

We clarified that this study examined the response of soil respiration to both drought and extreme high precipitation.

L 120: Replace "positive" and "negative" with something clearer. What is a "positive" response of a biome? Is it higher GPP, higher Respiration, higher NEP, higher biodiversity?

We have rephrased the sentence to clarify that extreme precipitation is likely to increase aboveground net primary productivity (ANPP) of xeric biomes and to reduce ANPP of mesic biomes.

L 139 to 143: This statement does not seem to be justified. Winter and spring are not key seasons for metabolic activity in irrigated croplands so the leading statement about smaller effects on the overall annual carbon balance seems to be misleading.

We have removed this statement.

L 143 to 144: "Combined..." This comment about the importance of timing and magnitude does not appear to be a synthesis statement, pertaining to only one study of those highlighted in the special issue.

L 145 to 147: "[However], extreme temperature events occurr[ing] in the growing season could substantially alter carbon fluxes, while those events occurr[ing] during ..."

L 145 to 147: This statement seems to correct or more correctly state the one above (L139 to 143).

These two sentences are synthesis statements. We have listed them as a separate paragraph to avoid confusion. "occurred" has been replaced with "occurring".

L 155: Include citation to: Ghimire B, Williams CA, Collatz GJ, Vanderhoof M, Rogan J, Kulakowski D, Masek JG (2015) "Large Carbon Release from Bark Beetle Outbreaks across Western United States Imposes Climate Feedback", Global Change Biology, doi: 10.11/gcb.12933.

This citation has been added.

L 159 to 160: clarify "benefit of herbivory to undamaged trees" and also, does this include understory non-tree species?

We have rephrased the sentence as follows: This study also indicates that the residual forest and the understory vegetation contributed to carbon uptake and could enable the forest to return to carbon neutrality at a faster rate than clear-cuts.

L 161 to 163: It seems the study highlighted here only looked at MPB and if so, how could it suggest that the impacts of herbivore outbreak depend on the type of herbivore?

This study (Mathys et al. 2013) only examined MPB. This study along with previous studies indicated that impacts of herbivore outbreak depend on the type of herbivore and the intensity of disturbance. We have made this clear in the revision.

L 166: It seems redundant to include NEP and "carbon exchange between the land and the atmosphere" given that NEP typically includes CO2 and that non CO2 carboncontaining molecules are rarely emphasized and do not seem to have been emphasized in the studies included in this special issue.

We have deleted "carbon exchange between the land and the atmosphere" to remove redundancy.

L 166: It might make sense to clarify what is meant by "subsequent changes in NEP" by noting the relevant processes such as respiration of disturbance-killed biomass, and any changes to net primary productivity.

We have clarified that the changes in NEP are due to changes in GPP and ecosystem respiration of the remaining live stand and the heterotrophic respiration of the damaged biomass.

L 173 to 174: Check the units on your trend, which should be Tg C yr-1 yr-1... 8 to 18 Tg C yr-1 is pretty big. Should this be over an interval of time?

The units are correct. This is a global-scale study. The rate of decreasing net carbon balance before the 1970s of the  $20^{\text{th}}$  century was estimated to be 8 Tg C yr-1, and the increasing rate was 18 Tg C yr-1 during the remainder of the  $20^{\text{th}}$  century.

L 207: The geographic domain of the Zhou et al. study should be reported. Was it global? Was it in North America or Europe? The Amazon? The quantitative figures reported must be region specific.

This is a global-scale synthesis study. However, the sites are mainly distributed in North America and Europe. We have described this in the revision.

L 220 to 223: this statement is very general and does not offer much in the way of findings. We have rephrased the statement.

L 222: "vulnerable" seems to be an odd term. All forests would be vulnerable only some are targeted because of economic value and modes of production.

We have rephrased this statement.

L 224: This heading "time since disturbance" does not appear to be a good fit for the studies highlighted below. You might think about a different heading / grouping. We have changed the heading to "Disturbance legacy".

L 228: "near the site" is vague and unclear. We have changed "near the site" to "at the site".

L 230: This paper does not seem to belong under the heading "Time since disturbance". Can it be better linked to the flow of the preview?

This paper examines vegetation recovery following fire disturbance and thus fits into this section.

L 232: Replace "found" with "supported the notion that". This is not a new finding, really, and is model based, so it seems somewhat out of place to state that it was "found". We replaced "found" with "supported the notion that".

L 234: Maybe connect these sentences... "carbon sink conditions, highlighting the importance of ..."

We did not combine these two sentences because of the length of the combined sentence.

L 239 to 245: suggest cutting this paragraph. It seems out of place and is redundant with things already mentioned elsewhere, including an earlier highlight of the Wang et al. 2014 study. It has a discussion of its own with citations to works outside of the scope of the special issue and thus seems out of place.

This paragraph has been removed as suggested.

L 252: which two? Wang et al. is not described as supporting this statement, so the statement seems to apply only to the Bond-Lamberty et al. 2015 study. We have clarified that this statement only applies to Bond-Lamberty et al. 2015.

L 256: Should there be a new heading here? Maybe "Challenges and Opportunities"? This paragraph does not include discussion on extreme climate events, and therefore we have decided to keep it as a part of the *Disturbance legacy* sub-section.

L 265: "conforming" to "confirming"

L 274; "will likely help" to "are helping to", and cite (e.g. Williams et al. 2014). Williams CA, Collatz GJ, Masek J, Huang C, Goward S (2014) "Impacts of disturbance history on forest carbon stocks and fluxes: Merging satellite disturbance mapping with forest inventory data in a carbon cycle model framework", Remote Sensing of Environment, 151:57-71, http://dx.doi.org/10.1016/j.rse.2013.10.034.

These changes have been made.

**Response to Referee #2's comments**

In preparing this revision, we have fully considered the reviewer (Dr. M. van Oijen)' comments and have revised the manuscript accordingly.

This paper provides an introduction to a special issue of Biogeochemistry. The issue consists of 17 papers on the impact of extreme climatic events and disturbances on ecosystem carbon dynamics. Fifteen of the studies are on terrestrial ecosystems, one study is on mangroves and one on lakes. The papers differ strongly in their choice of ecosystem, research question and methodology. That raises the question: what is the purpose of providing an introduction to such a heterogeneous collection? The authors show (Fig. 2) that more than 200 papers are now being published each year on the response of carbon dynamics to extreme events and disturbances, so why do the 17 issue papers merit special attention? The obvious justification for such an introduction is that it provides an opportunity to place recent papers in context, i.e. review the state of the art and identify remaining research gaps. This is attempted in the paper but could be done more systematically, as discussed in the following.

We thank the reviewer for the constructive comments on our preface. We have revised the manuscript as suggested.

The first section of the paper ("Introduction") can be summarised as follows: (1) Interannual variability (IAV) of GPP is especially large in the tropics, (2) extreme events and disturbances can affect carbon dynamics and will become more frequent and intense in the future, (3) their impacts can be studied with measurements and models - and many papers are being published; the ones here were the outcome from AGU-sessions.

This is fine as far as it goes, with the possible exception of the text on IAV measured as multiannual standard deviation of GPP (Fig. 1), which seems an unnecessary distraction. Extreme events are outliers, not standard deviations, and occur worldwide - and not just in the tropics.

We have replaced the interannual variability (IAV) map measured by standard deviation with the map measured by the coefficient of variation (CV; CV=standard deviation/mean). CV betters measures the IAV of carbon fluxes. We have also added a figure of the number of extreme annual values (i.e., outliers) (listed as Fig. 2). We identified the outliers on a per-pixel basis using the Boxplot concept. An outlier is defined as an annual GPP value that is either larger than the 75% quartile+1.5\*interquartile range or smaller than the 25% quartile – 1.5 \* interquartile range.

A brief analysis of the existing literature could be added to this introductory section, flagging up both what research has been done and what has not been done. Which ecosystem types, which extreme event types, which disturbances, at which locations and spatiotemporal scales have been studied so far, and which were overlooked? And which are likely to become more important in the future? Can we distinguish direct and indirect effects of disturbances and extreme events both on the same location and elsewhere? Do the 17 papers address any of the research gaps? I think the paper would gain from being more comprehensive and analytical - otherwise there is no added value compared to the special issue papers themselves. I understand that you want to keep the preface short, but you could delete the IAV-text and figure, replacing it with say 20-30 lines on the state of the art.

We agree that an overview of the literature would be of interest but, as noted in Figure 3 (originally listed as Figure 2), a large number of manuscripts have been published on the topic of

'extreme events' and on 'disturbances'. Any brief summary of this body of literature would barely do it justice. We feel that in this instance, a full review paper may be able to adequately (but probably not comprehensively) synthesize existing literature. We highlight the novel findings of the manuscripts - this is designed to communicate the gaps in knowledge addressed by each. Instead, we improved Figure 1 to also include an outlier analysis as an alternate approach for identifying regions of the globe that are prone to annual GPP values that exceed the normal statistical range.

The final section ("Conclusions") states future research needs. Three topics are men-tioned: (1) studying interactions between extreme events and disturbances, (2) collecting more data on disturbances, (3) improving models for disturbances. Whilst these certainly constitute worthwhile efforts, they seem an arbitrary and small selection of topics; many others could have been mentioned. And are there no research needs left for extreme events rather than disturbances? Also, there is no discernible relationship between the three listed research needs and the 17 papers of the special issue, so it remains unclear what the papers collectively have contributed. For example, at least five of the 17 papers used models: if those models still need to be improved, does that disqualify their current results?

We have revised this section by adding research needs for extreme events. The need for further improve does not disqualify the current results of these models but indicates our understanding of the underlying mechanisms of extreme climate events and disturbances and their representation in models are still limited.

**SPECIFIC COMMENTS**

1. 36-37: The Introduction begins with discussion of "terrestrial biosphere" and "Terrestrial carbon fluxes". This suggests that the special issue only considers terrestrial ecosystems, which is not the case. Begin by setting the scene (what kind of studies are being introduced by you) before delving into details like the IAV.

We have removed "terrestrial", and changed "terrestrial carbon fluxes" to "biospheric carbon fluxes".

1. 41: Add a reference to the MODIS work.

We have added a reference for the MODIS data product (Zhao et al. 2005).

1. 99-100: Those reductions of 28 and 38% are for which period? During the event, the year following the event, : : :?

The drought reduced GPP and carbon sink by 28% and 38% in the drought year – 2012. We have clarified this.

1. 172-174: What happened around 1970 that caused the trend break?

1. 215-216: Here you explain what partial cutting is, after having discussed the impacts of it already on l. 207.

We have moved the explanation of partial cutting to where the phrase first appeared in the paragraph.

1. 231-238: This section seems to ignore the current understanding that it is increased N-deposition, not elevated CO2, that has increased forest sink strength.

We have explicitly mentioned that N deposition – a factor enhancing ecosystem carbon uptake was not explicitly considered, although the effects of nitrogen deposition carbon sink strength have been controversial (Magnani et al., 2007; Nadelhoffer et al., 1999).

1. 252: Which two studies?

We have clarified what this study only applies to Bond-Lamberty et al. 2015.

TECHNICAL CORRECTIONS

24: Missing space after "by".
 28: Remove "layers".
 We have made these changes.

1. 42: "the Amazon" should be "Amazonia".

- 1. 43, 45, 46: Remove "on the order".
- 1. 50-51: Remove "terrestrial"?

1. 59-61: "We can only : : : scales" can safely be removed.

- 1. 65: Replace "mechanistic responses" by "mechanisms underlying responses".
- 1. 71: Add "the" before "consequences".

1. 76: The total number increases by 200 articles per year, not 20. Replace "total" with "annual". We have made these changes.

1. 82: AGU meeting: in which year(s)? We have clarified that the AGU meeting was in 2011-2013.

1. 85-87: "We feel : : : change" can be removed. We have retained this sentence as part of our evaluation of the authors' contribution..

1. 93-96: "That being said, : : : 2008": more waffling, please remove.

- 1. 110: Replace "have" with "has".
- 1. 145-146: replace "occurred" with "occurring".
- 1. 179: Remove "potential".
- 1. 182: Replace first dash with a space.
- 1. 201: Write "hurricanes".
- 1. 203: Remove "annual".

1. 256: Why write "data layers" instead of simply "data"? There is some GIS-jargon here (including the "polygons" of line 259 and two further "layers" in lines 290 and 292).

1. 260: "source of information".

1. 265: What does "conforming" mean?

1. 293: Remove "systematically".

These changes have been made as suggested.

**Preface: Impacts of extreme climate events and 1 disturbances on carbon dynamics 2 3 Jingfeng Xiao1, Shuguang Liu2, and Paul C. Stoy3,4 4 5 1Earth Systems Research Center, Institute for the Study of Earth, Oceans, and Space, University of New Hampshire, Durham, NH 03824, USA 6 7 2U.S. Geological Survey (USGS) Earth Resources Observation and Science (EROS) Center, Sioux Falls, SD 57198, USA 8 9 3Department of Land Resources and Environmental Sciences, Montana State University, 10 Bozeman, MT 59717, USA 4Institute on Ecosystems, Montana State University, Bozeman, MT 59717, USA 11 12 13 Correspondence to: J. Xiao (j.xiao@unh.edu) 14 15 Abstract The impacts of disturbances and extreme climate events and disturbances (ECE&D) 16 on the carbon cycle have received growing attention in recent years, as evidenced by the 17 increasing number of journal articles published on these topics. This special issue showcases a 18 collection of recent advancesadvancements in understanding the impacts of ECE&D on 19 disturbances and extreme events on the carbon cyclingcycle. Notable advancesadvancements include, but are not limited to, quantifying how harvesting activities impact forest structure, 20**

- 21 carbon pool dynamics, and recovery processes; observed drastic increases of the
  - 1

22 concentrations of dissolved organic carbon and dissolved methane in thermokarst lakes in western Siberia duringin a summer warming event; disentangling the roles of herbivores and 23 24 fire on forest carbon dioxide flux; direct and indirect impacts of fire on the global carbon balance; and improved atmospheric inversion of regional carbon sources and sinks by 25 26 incorporatingbyincorporating disturbances. Combined, studies herein indicate several major 27 research needs. First, disturbances and extreme events can interact with one another, and it is 28 important to understand their overall impacts and also disentangle their relative effects on the 29 carbon cycle. Second, current ecosystem models are not skillful enough to correctly simulate 30 the underlying processes and impacts of ECE&D (e.g., tree mortality and carbon 31 consequences). Third, benchmark data layers characterizing the timing, location, type, and 32 magnitude of disturbances must be systematically created to improve our ability to quantify 33 carbon dynamics over large areas. Finally, improving the representation of ECE&D in 34 regional climate/earth system models and accounting for the resulting feedbacks to climate 35 are essential for understanding the interactions between climate and ecosystem dynamics. Third, current ecosystem models are not skillful enough to correctly simulate the 36 37 impacts of disturbances such as disturbance-induced tree mortality and its carbon 38 consequences, and therefore must be improved to correctly represent underlying processes 39 and impacts.

40

**41 1 Introduction**

The terrestrial biosphere plays an important role in regulating atmospheric carbon dioxide concentrations and thereby climate. ExtremeTerrestrial carbon fluxes often exhibit pronounced interannual variability (IAV), and disturbances and extreme climate events such as droughtare primary sources of IAV (Eimers et al., 2008; Reichstein et al., 2013; Xiao et al., Paul Stoy 5/15/2016 10:52 AM Formatted: Don't adjust space between

2

Latin and Asian text, Don't adjust space between Asian text and numbers

[revised manuscript text omitted]

- Faw U, K. L., Schning, H. F., Scott, K. L., Suit, G., Suyker, A. E., and Torri, M. S., Estimation

- 596 of net ecosystem carbon exchange for the conterminous United States by combining MODIS 597 and AmeriFlux data, Agric. For. Meteorol., 148, 1827-1847,
- 598 10.1016/j.agrformet.2008.06.015, 2008.
- 599 Xiao, J. F., Zhuang, Q. L., Law, B. E., Baldocchi, D. D., Chen, J. Q., Richardson, A. D.,
- 600 Melillo, J. M., Davis, K. J., Hollinger, D. Y., Wharton, S., Oren, R., Noormets, A., Fischer,
- 601 M. L., Verma, S. B., Cook, D. R., Sun, G., McNulty, S., Wofsy, S. C., Bolstad, P. V., Burns,
- 602 S. P., Curtis, P. S., Drake, B. G., Falk, M., Foster, D. R., Gu, L. H., Hadley, J. L., Katulk, G.
- 603 G., Litvak, M., Ma, S. Y., Martinz, T. A., Matamala, R., Meyers, T. P., Monson, R. K.,
- Munger, J. W., Oechel, W. C., Paw U, K. T., Schmid, H. P., Scott, R. L., Starr, G., Suyker, A.
   E., and Torn, M. S.: Assessing net ecosystem carbon exchange of U.S. terrestrial ecosystems
- by integrating eddy covariance flux measurements and satellite observations, Agric. For.
- 607 Meteorol., 151, 60-69, 10.1016/j.agrformet.2010.09.002, 2011.
- Kiao, J. F., Zhuang, Q. L., Liang, E. Y., McGuire, A. D., Moody, A., Kicklighter, D. W.,
  Shao, X. M., and Melillo, J. M.: Twentieth-century droughts and their impacts on terrestrial
  carbon cycling in China, Earth Interact., 13, 10.1175/2009ei275.1, 2009.
- Yue, C., Ciais, P., Luyssaert, S., Cadule, P., Harden, J., Randerson, J., Bellassen, V., Wang,
  T., Piao, S. L., Poulter, B., and Viovy, N.: Simulating boreal forest carbon dynamics after
- stand-replacing fire disturbance: insights from a global process-based vegetation model,
  Biogeosciences, 10, 8233-8252, 10.5194/bg-10-8233-2013, 2013.
- Zeppel, M. J. B., Wilks, J. V., and Lewis, J. D.: Impacts of extreme precipitation and seasonal
  changes in precipitation on plants, Biogeosciences, 11, 3083-3093, 10.5194/bg-11-3083-2014,
- 617 2014.
- Zhao, M. S. and Running, S. W.: Drought-induced reduction in global terrestrial net primary
  production from 2000 through 2009, Science, 329, 940-943, 10.1126/science.1192666, 2010.
- Zhou, D., Zhao, S. Q., Liu, S., and Oeding, J.: A meta-analysis on the impacts of partial
  cutting on forest structure and carbon storage, Biogeosciences, 10, 3691-3703, 10.5194/bg10-3691-2013, 2013a.
- Zhou, D. C., Liu, S. G., Oeding, J., and Zhao, S. Q.: Forest cutting and impacts on carbon in
  the eastern United States, Scientific Reports, 3, 10.1038/srep03547, 2013b.
- 625 Zhu, Z. and Woodcock, C. E.: Continuous change detection and classification of land cover
- 626 using all available Landsat data, Remote Sens. Environ., 144, 152-171, 627 10.1016/j.rse.2014.01.011, 2014.
- 628
- 629